# A Head Formulation for the Steady-State Analysis of Water Distribution Systems Using an Explicit and Exact Expression of the Colebrook–White Equation

Mengning Qiu † and Avi Ostfeld *

Faculty of Civil and Environmental Engineering, Technion-Israel Institute of Technology, Haifa 32000, Israel;
* Correspondence: ostfeld@technion.ac.il
† Passed away.

**Abstract:** Steady-state demand-driven water distribution system (WDS) solution is the bedrock for much research conducted in the field related to WDSs. WDSs are modeled using the Darcy–Weisbach equation with the Swamee–Jain equation. However, the Swamee–Jain equation approximates the Colebrook–White equation, errors of which are within 1% for $\epsilon/D \in [10^{-6}, 10^{-2}]$ and $Re \in [5000, 10^8]$. A formulation is presented for the solution of WDSs using the Colebrook–White equation. The correctness and efficacy of the head formulation have been demonstrated by applying it to six WDSs with the number of pipes ranges from 454 to 157,044 and the number of nodes ranges from 443 to 150,630. The addition of a physically and fundamentally more accurate WDS solution method can improve the quality of the results achieved in both academic research and industrial application, such as contamination source identification, water hammer analysis, WDS network calibration, sensor placement, and least-cost design and operation of WDSs.

**Keywords:** water distribution system; demand-dependent models; head formulation; colebrook–white equation

## 1. Introduction

Water distribution systems (WDSs) are essential infrastructures of every city and town, the purpose of which is to satisfy the water requirements of the population, of agriculture and for industry with the required quality and quantity. The hydraulic steady-state solutions of WDSs (the flows and heads) is the foundation of many, if not all, WDS academic research and industry application. Therefore, the speed and accuracy of the hydraulic simulation model that is used to find the steady-state of WDSs underpin the quality of the research outputs and industry applications. The quest for a solution method for finding the steady-state solution of a looped WDS can date back to 1936. Since then, the research community diverged into three main branches: (1) loop-based methods, (2) null-space method, and (3) range space methods.

The loop-based methods use the loop energy equations and continuity equations to model the demand-driven steady-state of WDSs. The first loop-based methods and the first WDS solution method is the Hardy–Cross method [1]. The Hardy–Cross method is an iterative manual method that uses successive approximation to solve the above nonlinear system of equations in which a set of initial flows that satisfies the mass conservation equations is successively corrected until the stopping test has been met. The loop identification and the requirement that the initial guess of flows must satisfy continuity are the factors that affect the performance of the Hardy–Cross method. The performance of the loop-based methods is dependent on the sum of the length of all identified loop. The Shortest Cycle Basis [2] is the best set of loops that minimizes the time that is required to execute any loop-based method. This is because the use of the shortest cycle basis can achieve the minimum number of non-zeros in the key matrix. Ref. [3] explored the use of

minimum loops in the Newton–Raphson loop flows method and showed the time used to identify the shortest cycle basis is 100 to 10,000 times the time used to solve the network. Ref. [4] proposed two algorithms to select a set of network loops to achieve a highly sparse matrix. Although a smaller number of non-zeros in the Schur complement was reported in Creaco and Franchini [3], the substantial improvement in terms of the efficiency reported by Alvarruiz et al. [4] suggests the latter algorithm is the better practical choice. The overhead to identify the shortest cycle basis is clearly the bottleneck in the loop-based method even though the loop identification algorithm is only required to execute once for a given network topology.

The null-space methods, methods that operate in the subspace defined by a null space that is orthogonal to the column of the unknown-head node-arc incidences matrix, partitions the network into a spanning tree acyclic graph and the complementary chord tree edges. The addition of any chord tree pipe to the spanning tree graph creates a loop. The null-space method, in the context of WDS solution methods, is a special case of the loop-based methods. This is because the cycle basis created by any spanning tree graph and chord edges is a subset of the set of all cycle bases of the loop-based method, whereas the cycle basis of the loop-based method (particularly the shortest cycle basis) cannot be expressed as a combination of a spanning tree graph and the complementary chord tree edges. By sacrificing the generality, the null-space method requires fewer computation resources to find a combination of spanning tree graph and chord tree edges. Rahal [5] proposed a co-tree flows formulation in which it is necessary to (1) identify the associated circulating graph; (2) determine the demands that are to be carried by the spanning tree branches; (3) find the associated chain of branches closing a circuit for each co-tree chord; and (4) compute pseudo link head losses. Later, Elhay et al. [6] exploits the relationship between the co-tree flows and spanning tree flow by applying the Schilders' factorization [7] to permute the A1 matrix into a lower triangular square block at the top, representing a spanning tree, and a rectangular block below, representing the corresponding co-tree.

The range space methods include a collection of methods that operate in the subspace defined by the rows of the unknown-head node-arc incidences matrix. The global gradient algorithm [8] is the most widely used WDS solution method, which solves for the flows and the heads in WDSs simultaneously by exploiting the block structure of the Jacobian matrix to reduce the size of the key matrix in the linearization of the Newton method. The graph matrix partitioning algorithm (GMPA) [9] exploits the linear relationship between the flows in the internal trees and the flows in the superlinks to speed up the solution process of the GGA.

In addition to the different solution methods developed for the simulation of the WDS steady-state, network partitioning using graph theory has been another active research avenue. Network partitioning identifies sections of a WDS network that are hydraulically independent. The first graph structure that is exploited is the forest component of a WDS. A tree in a graph is any two vertices are connected by exactly one edge. Most WDSs have trees, the collections of which are called forests. Simpson et al. [10] proposed a forest-core partitioning algorithm to partition a WDS graph into its linear forest component and nonlinear core component. The flows in the forest pipes can be computed a priori and the heads in the forest nodes can be computed a posteriori by a linear process. Qiu et al. [11] proposed a bridge-block partitioning algorithm to partition a WDS graph into several linear independent blocks and bridges. The steady-state solution of one block of a WDS can be computed independently from other WDS blocks.

EPANET2 [12] is one of the most widely used WDS simulation packages, in which the GGA is used to provide steady-state demand-driven solution of WDSs. The code for EPANET 2 is in the public domain, allowing many studies to be conducted. These include the least-cost design and operation of WDSs, sensor placement, chlorine decay models calibration, contamination event detection, network vulnerability analysis, cyber-attack detection, network decontamination, and many others. Any inaccuracies that existed in the WDS solution method used will be inherited and sometimes exacerbated. One of the

accuracy-related problems is the use of the Swamee–Jain equation [13], an approximation of the implicit Colebrook–White equation [14], a general equation that can be used when the Reynolds number is greater than or equal to 4000, to calculate the friction factor. It is reported in Swamee and Jain [13] that the errors involved in friction factor are within 1% for $\epsilon/D \in [10^{-6}, 10^{-2}]$ and when $Re \in [5000, 10^8]$. All WDS solution methods that are currently available (1) ignored the above 1% error and (2) extended the applicability of the Swamee–Jain equation to calculate the friction factor when $Re \geq 4000$ and all values of $\epsilon/D$. This is mainly due to the solutions of the inexplicit Colebrook–White equation are required many times for multiple iterations, which is time-consuming.

In this paper, an iterative solution method that finds the steady-state demand-driven WDS solution using the Colebrook–White equation for the turbulent flow regime is proposed. This is achieved by (1) expressing the Colebrook–White equation, an implicit function, as an explicit, exact, and differentiable function to describe the friction factor in the turbulent flow regime, (2) using the Hagen-Poiseuille equation to describe the friction factor in the laminar flow regime, and (3) using cubic interpolation to fit a curve between the turbulent and the laminar flow regimes to describe the transitional flow regime. It is important to note that unlike the Swamee–Jain equation, the explicit expression of the Colebrook–White equation agree at all points, the use of which eliminates the inaccuracy that is associated with the Swamee–Jain equation. It is shown in this paper that the different steady-state solutions of WDSs are observed which can be a critical problem for some research areas (such as the least-cost design and operation, contamination event detection, network vulnerability analysis, sensor placement, district meter area, etc.).

This paper is organized as follows. A description of the general WDS demand-driven steady-state problem is given in the next section. This is followed by Section 3 the derivation of the proposed head formulation. Section 4 gives an algorithmic description of the proposed head formulation, followed by the validation of the proposed friction factor equation against the Colebrook–White equation. Six case study networks are then described in Section 6, the results of which are discussed in the next section. The last section offers some conclusions.

## 2. General WDS Demand-Driven Steady-State Problem

### 2.1. Definitions and Notation

Consider a water distribution system that contains $n_p$ pipes, $n_j$ junctions, and $n_r$ fixed-head nodes.

The $i$-th node of the network has two properties: its nodal demand $d_i$ and its elevation head $z_i$. Let $\boldsymbol{h} = \left( h_1, h_2, \ldots, h_{n_j} \right)^T$ denote the vector of unknown heads, $\boldsymbol{d} = \left( d_1, d_2, \ldots, d_{n_j} \right)^T$ denote the vector of nodal demands, $\boldsymbol{e_l} = \left( e_{l_1}, e_{l_2} \ldots, e_{l_{n_r}} \right)^T$ denote the vector of fixed-head elevations.

The $p$-th pipe of the network can be characterized by its diameter $D_p$, length $L_p$, flows $q_p$, and Hazen–William coefficient $C_p$ for Hazen–William head loss model or roughness height $\epsilon_p$ for Darcy–Weisbach head loss model. Let $\boldsymbol{q} = \left( q_1, q_2, \ldots, q_{n_p} \right)^T$ denote the vector of unknown flows, $\boldsymbol{C} = \left( C_1, C_2, \ldots, C_{n_p} \right)^T$ denote the vector of Hazen–William coefficients, $\boldsymbol{\epsilon} = \left( \epsilon_1, \epsilon_2, \ldots, \epsilon_{n_p} \right)^T$ denote the vector of roughness heights, $\boldsymbol{L} = \left( L_1, L_2, \ldots, L_{n_p} \right)^T$ denote the vector of pipe lengths, $\boldsymbol{D} = \left( D_1, D_2, \ldots, D_{n_p} \right)^T$ denote the vector of pipe diameters,

The matrix $\boldsymbol{A_1}$ is the full-rank, unknown-head, node-arc incidence matrix. The matrix $\boldsymbol{A_2}$ is the fixed-head node-arc incidence matrix. The head loss exponent $n$ is assumed to be dependent only on the head loss model: $n = 2$ for the Darcy–Weisbach head loss model and $n = 1.852$ for the Hazen-Williams head loss model. The head loss within the pipe $p$ is modeled by $h_{f_p} = r_p q_p |q_p|^{n-1}$. Denote by $\boldsymbol{G}(\boldsymbol{q}) \in \mathbb{R}^{n_p \times n_p}$, a diagonal square matrix with

elements $[\boldsymbol{G}]_{pp} = r_p |q_p|^{n-1}$ for $j = 1, 2, \ldots n_p$. Denote by $\boldsymbol{F}(\boldsymbol{q}) \in \mathbb{R}^{n_p \times n_p}$, a diagonal square matrix where the $p$-th element on its diagonal $[\boldsymbol{F}]_{pp} = \frac{d}{dq_p} [\boldsymbol{G}]_{pp} q_p$.

### 2.2. System of Equations

The steady-state flows and heads in a WDS system are modeled by the demand-driven model (DDM) continuity Equation (1) and the energy conservation Equation (2):

$$- \boldsymbol{A_1}^T \boldsymbol{q} - \boldsymbol{d} = \boldsymbol{O} \tag{1}$$

$$\boldsymbol{h}_f - \boldsymbol{A_1}\boldsymbol{h} - \boldsymbol{A_2}\boldsymbol{e}_l = \boldsymbol{O}, \tag{2}$$

which can be expressed as

$$\begin{pmatrix} \boldsymbol{G}(\boldsymbol{q}) & -\boldsymbol{A_1} \\ -\boldsymbol{A_1}^T & \boldsymbol{O} \end{pmatrix} \begin{pmatrix} \boldsymbol{q} \\ \boldsymbol{h} \end{pmatrix} - \begin{pmatrix} \boldsymbol{A_2}\boldsymbol{e}_l \\ \boldsymbol{d} \end{pmatrix} = 0, \tag{3}$$

where its Jacobian matrix used in the solution process is

$$\boldsymbol{J} = \begin{pmatrix} \boldsymbol{F}(\boldsymbol{q}) & -\boldsymbol{A_1} \\ -\boldsymbol{A_1}^T & \boldsymbol{O} \end{pmatrix} \tag{4}$$

and it is sometimes referred to as a nonlinear saddle point problem [15].

This nonlinear system is often solved by the Newton method, in which $\boldsymbol{q}^{(m+1)}$ and $\boldsymbol{h}^{(m+1)}$ are repeatedly computed from $\boldsymbol{q}^{(m)}$ and $\boldsymbol{h}^{(m)}$ by

$$\begin{pmatrix} \boldsymbol{F}^{(m)}(\boldsymbol{q}^{(m)}) & -\boldsymbol{A_1} \\ -\boldsymbol{A_1}^T & \boldsymbol{O} \end{pmatrix} \begin{pmatrix} \boldsymbol{q}^{(m+1)} - \boldsymbol{q}^{(m)} \\ \boldsymbol{h}^{(m+1)} - \boldsymbol{h}^{(m)} \end{pmatrix} = - \begin{pmatrix} \boldsymbol{G}^{(m)}\boldsymbol{q}^{(m)} - \boldsymbol{A_1}\boldsymbol{h}^{(m)} - \boldsymbol{A_2}\boldsymbol{e}_l \\ -\boldsymbol{A_1}^T\boldsymbol{q}^{(m)} - \boldsymbol{d}, \end{pmatrix} \tag{5}$$

until the relative differences $\frac{||\boldsymbol{q}^{(m+1)} - \boldsymbol{q}^{(m)}||}{||\boldsymbol{q}^{(m+1)}||}$ and $\frac{||\boldsymbol{h}^{(m+1)} - \boldsymbol{h}^{(m)}||}{||\boldsymbol{h}^{(m+1)}||}$ are sufficiently small.

## 3. Derivation of the Head Formulation

Consider the continuity equations in Equation (1) and a vector of unknown heads, $\boldsymbol{h}$, for the $n_j$ nodes in a network.

Assume the flows in pipe $p$ can be expressed as a function of the head loss in pipe $p$:

$$q_p = \mathfrak{f}([h_f]_p). \tag{6}$$

Let $\boldsymbol{a}_{*,j} \in \mathbb{R}^{n_p \times 1}$ denote the $j$-th column of the $\boldsymbol{A_1}$ matrix, the continuity equation can be rewritten as:

$$\boldsymbol{f}_c = \begin{pmatrix} f_{c_1} \\ f_{c_2} \\ \vdots \\ f_{c_j} \\ \vdots \\ f_{c_{n_j}} \end{pmatrix} = \begin{pmatrix} \boldsymbol{a}_{*,1}^T\boldsymbol{q} + d_1 \\ \boldsymbol{a}_{*,2}^T\boldsymbol{q} + d_2 \\ \vdots \\ \boldsymbol{a}_{*,j}^T\boldsymbol{q} + d_j \\ \vdots \\ \boldsymbol{a}_{*,n_j}^T\boldsymbol{q} + d_{n_j} \end{pmatrix} = \begin{pmatrix} \boldsymbol{a}_{*,1}^T\mathfrak{f}(h_f) + d_1 \\ \boldsymbol{a}_{*,2}^T\mathfrak{f}(h_f) + d_2 \\ \vdots \\ \boldsymbol{a}_{*,j}^T\mathfrak{f}(h_f) + d_j \\ \vdots \\ \boldsymbol{a}_{*,n_j}^T\mathfrak{f}(h_f) + d_{n_j} \end{pmatrix}$$

is the mass balance equation expressed for every node in a WDS.

The partial derivative of $f_c(q_1, q_2, \ldots, q_{n_p})$ with respect to $h_f$ can be expressed as:

$$J = \frac{\partial f_c}{\partial h_f} = \begin{pmatrix} \frac{\partial f_{c_1}}{\partial h_1} & \frac{\partial f_{c_1}}{\partial h_2} & \cdots & \frac{\partial f_{c_1}}{\partial h_{n_j}} \\ \frac{\partial f_{c_2}}{\partial h_1} & \frac{\partial f_{c_2}}{\partial h_2} & \cdots & \frac{\partial f_{c_2}}{\partial h_{n_j}} \\ \vdots & \vdots & \ddots & \vdots \\ \frac{\partial f_{c_{n_j}}}{\partial h_1} & \frac{\partial f_{c_{n_j}}}{\partial h_2} & \cdots & \frac{\partial f_{c_{n_j}}}{\partial h_{n_j}} \end{pmatrix} \tag{7}$$

in which the partial derivative of $f_{c_j}(q_1, q_2, \ldots, q_{n_p})$ with respect to $h_{f_p}$ can be expressed as:

$$\frac{\partial f_{c_j}}{\partial h_j} = \sum_{i=1}^{n_p} \left( \frac{\partial f_{c_j}}{\partial q_i} \frac{\partial q_i}{\partial h_j} \right). \tag{8}$$

As a result, Equation (7) can be written as:

$$J = \begin{pmatrix} \frac{\partial f_{c_1}}{\partial q_1} & \frac{\partial f_{c_1}}{\partial q_2} & \cdots & \frac{\partial f_{c_1}}{\partial q_{n_p}} \\ \frac{\partial f_{c_2}}{\partial q_1} & \frac{\partial f_{c_2}}{\partial q_2} & \cdots & \frac{\partial f_{c_2}}{\partial q_{n_p}} \\ \vdots & \vdots & \ddots & \vdots \\ \frac{\partial f_{c_{n_j}}}{\partial q_1} & \frac{\partial f_{c_{n_j}}}{\partial q_2} & \cdots & \frac{\partial f_{c_{n_j}}}{\partial q_{n_p}} \end{pmatrix} \begin{pmatrix} \frac{\partial q_1}{\partial h_1} & \frac{\partial q_1}{\partial h_2} & \cdots & \frac{\partial q_1}{\partial h_{n_j}} \\ \frac{\partial q_2}{\partial h_1} & \frac{\partial q_2}{\partial h_2} & \cdots & \frac{\partial q_2}{\partial h_{n_j}} \\ \vdots & \vdots & \ddots & \vdots \\ \frac{\partial q_{n_p}}{\partial h_1} & \frac{\partial q_{n_p}}{\partial h_2} & \cdots & \frac{\partial q_{n_p}}{\partial h_{n_j}} \end{pmatrix} + \begin{pmatrix} \frac{\partial d_1}{\partial h_1} & 0 & \cdots & 0 \\ 0 & \frac{\partial d_2}{\partial h_2} & \cdots & 0 \\ \vdots & \vdots & \ddots & \vdots \\ 0 & 0 & \cdots & \frac{\partial d_{n_j}}{\partial h_{n)j}} \end{pmatrix} \tag{9}$$

where the first matrix of the matrix multiplication above can be written as $\partial A_1^T q / \partial q$, which is $A_1^T$, and the partial derivative of $q_p(h_{f_p})$ (Equation (6)) with respect to $h_j$ can be expressed as:

$$\frac{\partial q_p}{\partial h_j} = \frac{\partial q_p}{\partial h_{f_p}} \frac{\partial h_{f_p}}{\partial h_j}. \tag{10}$$

Substitute Equation (10) into Equation (9), we get:

$$J = A_1^T \begin{pmatrix} \frac{\partial q_1}{\partial h_{f_1}} \frac{\partial h_{f_1}}{\partial h_1} & \frac{\partial q_1}{\partial h_{f_1}} \frac{\partial h_{f_1}}{\partial h_2} & \cdots & \frac{\partial q_1}{\partial h_{f_1}} \frac{\partial h_{f_1}}{\partial h_{n_j}} \\ \frac{\partial q_2}{\partial h_{f_2}} \frac{\partial h_{f_2}}{\partial h_1} & \frac{\partial q_2}{\partial h_{f_2}} \frac{\partial h_{f_2}}{\partial h_2} & \cdots & \frac{\partial q_2}{\partial h_{f_2}} \frac{\partial h_{f_2}}{\partial h_{n_j}} \\ \vdots & \vdots & \ddots & \vdots \\ \frac{\partial q_{np}}{\partial h_{f_{np}}} \frac{\partial h_{f_{np}}}{\partial h_1} & \frac{\partial q_{np}}{\partial h_{f_{np}}} \frac{\partial h_{f_{np}}}{\partial h_2} & \cdots & \frac{\partial q_{np}}{\partial h_{f_{np}}} \frac{\partial h_{f_{np}}}{\partial h_{n_j}} \end{pmatrix} + \frac{\partial d}{\partial h}$$

$$= A_1^T \begin{pmatrix} \frac{\partial q_1}{\partial h_{f_1}} & 0 & \cdots & 0 \\ 0 & \frac{\partial q_2}{\partial h_{f_2}} & \cdots & 0 \\ \vdots & \vdots & \ddots & \vdots \\ 0 & 0 & \cdots & \frac{\partial q_{np}}{\partial h_{f_{np}}} \end{pmatrix} \begin{pmatrix} \frac{\partial h_{f_1}}{\partial h_1} & \frac{\partial h_{f_1}}{\partial h_2} & \cdots & \frac{\partial h_{f_1}}{\partial h_{n_j}} \\ \frac{\partial h_{f_2}}{\partial h_1} & \frac{\partial h_{f_2}}{\partial h_2} & \cdots & \frac{\partial h_{f_2}}{\partial h_{n_j}} \\ \vdots & \vdots & \ddots & \vdots \\ \frac{\partial h_{f_{np}}}{\partial h_1} & \frac{\partial h_{f_{np}}}{\partial h_2} & \cdots & \frac{\partial h_{f_{np}}}{\partial h_{n_j}} \end{pmatrix} + \frac{\partial d}{\partial h} \tag{11}$$

where the third matrix in the matrix multiplication above can be expressed as $(\partial A_1 h + A_2 e_l)/\partial h$, which is $A_1$. Denote by $W \in \mathbb{R}^{n_p \times n_p}$, a diagonal square matrix where the $p$-th element on its diagonal $[W]_{pp} = f'(h_{f_p})$. The Jacobian matrix in Equation (11) can be expressed as:

$$J = A_1^T W A_1 + \frac{\partial d}{\partial h} \tag{12}$$

Omitting term $\frac{\partial \boldsymbol{d}}{\partial \boldsymbol{h}}$ for the demand-driven analysis, this new WDS solution method can be solved by the Newton method in which $d\boldsymbol{h}^{(m+1)}$ are repeatedly computed from $\boldsymbol{h}^{(m)}$ using:

$$J\left(\boldsymbol{h}^{(m+1)} - \boldsymbol{h}^{(m)}\right) = -A_1^T \boldsymbol{q} - \boldsymbol{d} \tag{13}$$

Moving $\boldsymbol{h}^{(m)}$ to the right-hand-side of Equation (13), we get:

$$J\boldsymbol{h}^{(m+1)} = -A_1^T \boldsymbol{q}^{(m)} - \boldsymbol{d} + A_1^T \boldsymbol{W}\left(\boldsymbol{h}_f^{(m)} - A_2 \boldsymbol{e}_l\right) \tag{14}$$

Equation (6) is now derived for Hazen-Williams and Darcy–Weisbach head loss models in the next subsections.

### 3.1. Hazen-Williams Equation

The Hazen-Williams Equation head loss equation is

$$h_{f_p} = \frac{10.67 L_p}{C_p^{1.852} D_p^{4.8704}} q_p |q_p|^{0.852} \tag{15}$$

and can be rewritten as:

$$q_p = \mathfrak{f}_{HW}(h_{f_[}) = \frac{C_p D_p^{2.63}}{3.6 L_p^{0.54}} h_{f_p} |h_{f_p}|^{-0.46} \tag{16}$$

and the partial derivative of $q_p(h_{f_p})$ (Equation (16)) with respect to $h_{f_p}$ can be expressed as:

$$\mathfrak{f}'_{HW}(h_{f_p}) = \frac{0.54 C_p D_p^{2.63}}{3.6 L_p^{0.54}} |h_{f_p}|^{-0.46} \tag{17}$$

### 3.2. Darcy–Weisbach Equation

The Darcy–Weisbach head loss equation is:

$$h_{f_p} = \frac{8 f_p L_p}{\pi^2 g D_p^5} |q_p| q_p \tag{18}$$

Equation (18) can be rearranged into

$$f_p = \frac{\pi^2 g D_p^5}{8 L_p |q_p| q_p} h_{f_p} \tag{19}$$

The friction factor for turbulent flow regime ($Re \geq 4000$) can be calculated by using the Colebrook–White equation:

$$\frac{1}{\sqrt{f_p}} = -2\log\left(\frac{\epsilon_p}{3.7 D_p} + \frac{2.51}{Re_p \sqrt{f_p}}\right) \tag{20}$$

in which the Reynolds number is

$$Re_p = \frac{4|q_p|}{\pi D_p \nu}. \tag{21}$$

Substitute Equations (19) and (21) into the RHS of Equation (20), we get:

$$f_p = \frac{0.25}{\log^2\left(\frac{\epsilon_p}{3.7 D_p} + \frac{2.51\sqrt{2}\nu L_p^{0.5}}{2|h_{f_p}|^{0.5} g^{0.5} D_p^{1.5}}\right)} \tag{22}$$

Denote by $S_p = h_{f_p}/L_p$, the hydraulic gradient in pipe $p$. The friction factor in pipe $p$ can be expressed as:

$$f_p = \frac{0.25}{\log^2\left(\frac{\epsilon}{3.7D_p} + \frac{2.51\sqrt{2}\nu}{2g^{0.5}D_p^{1.5}}|S_p|^{-0.5}\right)} \tag{23}$$

The friction factor for laminar flow regime ($Re \leq 2000$) can be calculated by using the Hagen-Poiseuille equation:

$$|q_p| = \frac{16\pi D_p \nu}{f_p} \tag{24}$$

Substitute Equation (24) into Equation (18), we get:

$$f_p = \frac{2048\nu^2}{gD_p^3}|S_p|^{-1} \tag{25}$$

The friction factor for transitional flow regime ($2000 \leq Re \leq 4000$) can be calculated by using a cubic interpolation between the laminar flow regime and the turbulent flow regime using:

$$f = (c_0 + S(c_1 + S(c_2 + Sc_3))) \tag{26}$$

in which $S = (|S_p| - S_{p2000})/(S_{p4000} - S_{p2000})$, $c_0 = 0.032$, $c_1 = -0.032$, $c_2 = -0.032 + 3f_{p4000} - f'_{p4000}$, $c_3 = 0.032 - 2f_{p4000} + f'_{p4000}$, $f_{p4000}$ is the friction factor for pipe $p$ when the Reynolds number is 4000, and $f'_{p4000}$ is the derivative of friction factor when the Reynolds number is 4000.

Let $a_p = \frac{2.51\sqrt{2}\nu}{2g^{0.5}D_p^{1.5}}$, we can express:

$$f_p = \begin{cases} \frac{2048\nu^2}{gD_p^3}|S_p|^{-1} & |S_p| \leq S_{p2000} \\ (c_0 + S(c_1 + S(c_2 + Sc_3))) & S_{p2000} \leq |S_p| \leq S_{p4000} \\ \frac{0.25\ln^2(10)}{\ln^2\left(\frac{\epsilon}{3.7D} + a_p|S_p|^{-0.5}\right)} & |S_p| \geq S_{p4000} \end{cases} \tag{27}$$

Let $b_p = \frac{\pi g^{0.5}D_p^{2.5}}{2\sqrt{2}}$, the flows in pipe $p$ as a function of the head loss in pipe $p$ (Equation (6)) for the Darcy–Weisbach head loss model can be expressed as

$$q_p(S_p) = b_p\left(\frac{S_p}{|S_p|^{0.5}f_p^{0.5}}\right) \tag{28}$$

where its derivative can be expressed as:

$$q'(S_p) = f'_{DW}(S_p) = 0.5b_p(|S_p|f_p)^{-0.5}\left(1 - \frac{S_p}{f_p}\frac{df_p}{dS_p}\right) \tag{29}$$

in which

$$\frac{df_p}{dS_p} = \begin{cases} -\frac{2048\nu^2}{gD_p^3}(S_P|S_P|)^{-1} & |S_P| \leq S_{p2000} \\ \frac{c_1 + 2Sc_2 + 3S^2c_3}{(S_{4000} - S_{2000})}S_p|S_p|^{-1} & S_{p2000} \leq |S_p| \leq S_{p4000}, \\ \frac{a_p\ln^2(10)}{4\ln^3\left(\frac{\epsilon}{3.7D} + a_p|S_p|^{-0.5}\right)\left(\frac{\epsilon}{3.7D}|S_p|^{0.5} + a_p\right)S_p} & |S_p| \geq S_{p4000} \end{cases} \tag{30}$$

and $\frac{dS_p}{dh_{f_p}} = L^{-1}$. It is important to note that special care must be taken to calculate the $q'(h_f)$ value for laminar flows as the pipe head loss will be cancelled when calculating the pipe flow:

$$q(S_p) = b_p \left( \frac{2048v^2}{gD_p^3} \right)^{-0.5} S_p \tag{31}$$

as a result, the derivative of the pipe flow with respect to the pipe head loss in the laminar flow regime can be expressed:

$$q'(S_P) = b_p \left( \frac{2048v^2}{gD_p^3} \right)^{-0.5} \tag{32}$$

which is a constant value. Therefore, a pipe with zero head loss will not cause any numerical problems as a result of the zero head loss.

Finally, $S_{p2000}$ and $S_{p4000}$ need to be identified for pipe $p$ for all pipes. $S_{p2000}$ can be easily identified as $f_{2000} = 64/2000 = 0.032$ for any pipes and $q_{p2000} = 500 \, \pi D_p v$. Therefore,

$$S_{p2000} = \frac{64000v^2}{gD_p^3}. \tag{33}$$

The determination of $S_{p4000}$ involves the solution of $f_{p4000}$ using the implicit Cole-brook–White equation for pipe $p$ when $Re = 4000$ and substitute the value of $f_{p4000}$ in Equation (2). This is only time when the direct solution of the implicit Colebrook–White Equation is required.

## 4. Head Formulation Algorithm

The steps of the proposed head formulation are described in Algorithm 1. The proposed head formulation, a single-phase formulation, within the iterative phase (between lines 4 and 18 in Algorithm 1 is similar in terms of the computational intensive when compared to the global gradient algorithm, a two-phase formulation.

Meanwhile, the overhead of the proposed head formulation, particularly the computation of the $S_{p4000}$ that requires the solution of the Colebrook–White equation for the Darcy–Weisbach head loss model, can increase the computation burden of the algorithm. However, as the manufacturing limitation, there is only a limit number of pipes that is available for the construction of WDSs. Therefore, the number of combinations of the pipe roughness heights and pipe diameters is limited, so does the number of distinct values of $S_{p4000}$. This is also true for the least-cost design problem of a WDS. $S_{p4000}$ value is only required to be computed and stored for each of the commercially available pipes a priori. This stored value can then be retrieved during the optimization phase.

Moreover, Equation (16) is just an inverse function of Equation (15). As a result, the $\frac{dq}{dh_f}$ required in the proposed head formulation is the inverse of the $\frac{dh_f}{dq}$ required in the GGA. Therefore, the Schur Complement in the GGA, $A_1{}^T F^{-1} A_1$, is the same as the Jacobian matrix shown in Equation (12) as $W = F^{-1}$. Thus, if the proposed head formulation and the GGA starts with the same initial starting points when applied to a WDS with Hazen-Williams head loss model, they will also have the same iterative solutions and the same final solution.

---

**Algorithm 1:** Head formulation

**input**    : $L$: Pipe lengths; $D$: Pipe diameters; $\epsilon$: pipe roughness heights or $C$: Hazen-Williams coefficients; $e_I$: source elevation heads; $d$ nodal demands; $hdlss$: head loss model ($DW$ for Darcy–Weisbach and $HW$ for Hazen-Williams) .

**parameters**: $\nu$: water viscosity; g: gravitational constant

**output**   : $q$: Pipe flows; $h$ Nodal heads

1   $h_f^{(0)}$=rand($n_p$,1); **if** $hdlss$ =$DW$ **then**

2     Calculate $S_{p2000}$ using Equation (33);

3     Calculate $S_{p4000}$ using the Colebrook–White equation as in Equation (20);

4   **endif**

5   **for** *i=1:maxIter* **do**

6     **if** $hdlss$ =$DW$ **then**

7       Calculate $f_p^{(i-1)}$ using Equation (27);

8       Calculate $q^{(i-1)}$ using $\mathfrak{f}_{DW}$ in Equation (28);

9       Calculate $\dfrac{dq^{(i-1)}}{dh_f^{(i-1)}}$ using $\mathfrak{f}'_{DW}$ in Equation (29);`/* use Equation  (32) for`

         `the laminar flow regime                                    */`

10    **else**

11      Calculate $q^{(i-1)}$ using Equation (16);

12      Calculate $\dfrac{dq^{(i-1)}}{dh_f^{(i-1)}}$ using $\mathfrak{f}'_{DW}$ in Equation (17);

13    **endif**

14    Calculate $h^{(i)}$ using Equation (14);

15    **if** $\left\lVert \dfrac{\left(h^{(i)}-h^{(i-1)}\right)}{h^{(i)}} \right\rVert_\infty \leq tol$ **then**

16      **break**;

17    **endif**

18    Calculate $h_f^{(i)}$ using Equation (2);

19 **end for**

---

## 5. Validation of the Proposed Friction Factor Equation

The proposed friction factor equation is validated in this section. The proposed friction factor equation and the Swamee–Jain friction factor equation will be used to compute friction factors for pipes with the value of $\epsilon = (1\text{ mm}, 0.1\text{ mm}, 0.01\text{ mm}, 0.001\text{ mm})$, $D = 100$ mm and Reynolds number ranging from 4000 to $10^8$, and the computed friction factor value in turbulent flow regime will be plotted against the value of the friction factor computed by using the Colebrook–White equation. As can be seen from Figure 1, the differences are observed between the friction factors computed by using the Swamee–Jain friction factor equation and that computed by using the Colebrook–White equation as shown in Swamee and Jain [13], whereas the friction factors computed by using the proposed friction factor equation are the same as that computed by using the Colebrook–White equation.

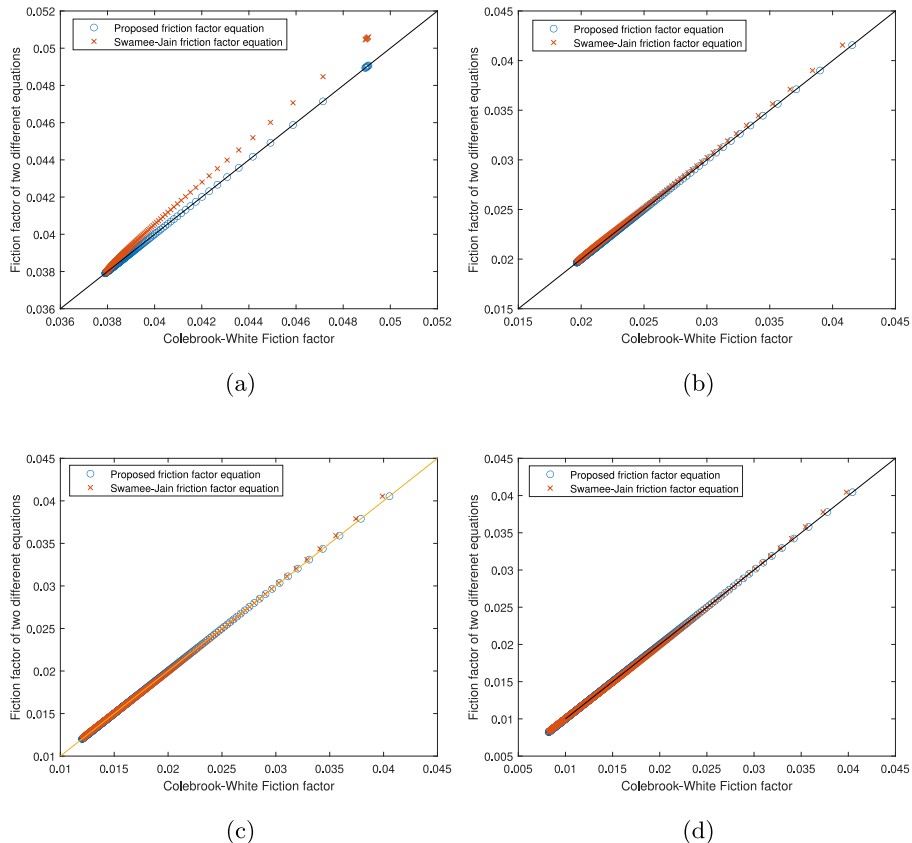

**Figure 1.** The friction factors calculated for the turbulent flow region using the proposed equation in Equation (27) and Swamee–Jain equation for: (**a**) $\epsilon/D = 10^{-2}$, (**b**) $\epsilon/D = 10^{-3}$, (**c**) $\epsilon/D = 10^{-4}$, and (**d**) $\epsilon/D = 10^{-5}$.

## 6. Case Studies

The proposed head formulation was implemented in WDSLib [16] and has been applied to six WDSs. The networks used here were assigned network identifiers from $N_1$ to $N_6$ (see details in Table 1):

- Network identifier $N_1$ is assigned to Balerma Network, first introduced by Reca and Martínez [17], is comprised of 454 pipes, 443 junctions, and four reservoirs.
- Network identifier $N_2$ is assigned to Richmond Network, first introduced by Van Zyl et al. [18], is comprised of 834 pipes, 848 junctions, and eight reservoirs.
- Network identifier $N_3$ is assigned to exnet Network, first introduced by Farmani et al. [19], is comprised of 2465 pipes, 1890 junctions, and three reservoirs. This is important to note that valves are replaced by pipes.
- Network identifier $N_4$ is assigned to the large Network used in Sitzenfrei et al. [20]. This network has 4021 pipes, 3557 junctions and one reservoir.
- Network identifier $N_5$ is assigned to Network 2 of the Battle of Network Sensors competition [21]. Network $N_5$ is comprised of 14,830 pipes, 12,523 junctions, and seven reservoirs. This is important to note that valves and pumps are replaced by pipes and demand patterns are removed.
- Network identifier $N_6$ is assigned to virtRome Network, first introduced by [20], is comprised of 157,044 pipes, 150,630 junctions, and four reservoirs.

All case studies were performed on an Intel Core i7-7700 running at 3.6 GHz with four cores in C++ under IEEE-standard double-precision floating arithmetic [22] with machine epsilon $\epsilon_{mach} = 2.204 \times 10^{-16}$.

**Table 1.** Benchmark networks summary.

| Network | No. of Pipes | No. of Nodes | No. of Sources | No. of Forest Pipes |
|---------|--------------|--------------|----------------|---------------------|
| $N_1$ | 454 | 443 | 4 | 288 |
| $N_2$ | 934 | 848 | 8 | 361 |
| $N_3$ | 2465 | 1890 | 3 | 429 |
| $N_4$ | 4021 | 3557 | 1 | 1566 |
| $N_5$ | 14,830 | 12,523 | 7 | 2932 |
| $N_6$ | 157,044 | 150,630 | 4 | 45,736 |

The infinity norm of the relative head differences $\left\| \frac{h^{m+1} - h^m}{h^{m+1}} \right\|_\infty$ will be used as convergence test. Without a WDS solution method that can be used as a benchmark, the friction factor residual in Equation (34), the continuity equation residual in Equation (35), and the energy equation residual in Equation (36) were used to validate the correctness of the proposed head formulation.

$$\sigma_f^{(m)} = \left\| \frac{1}{\sqrt{f^{(m)}}} + 2\log\left( \frac{\epsilon}{3.7D} + \frac{2.51}{Re^{(m)}\sqrt{f^{(m)}}} \right) \right\|_\infty \tag{34}$$

$$\sigma_c^{(m)} = \left\| A_1^T q^{(m)} + d^{(m)} \right\|_\infty \tag{35}$$

$$\sigma_E^{(m)} = \left\| h_f^{(m)} + A_1 h^{(m)} + A_2 e_l^{(m)} \right\|_\infty \tag{36}$$

## 7. Results and Discussion

Figure 2 shows the convergence of the proposed iterative head formulation. Networks $N_1$ and $N_2$ are both relatively small network, both of which have a small number of loops (11 loops for network $N_1$ and 14 loops for network $N_2$). The stopping test and the continuity residual have been met for both smaller size networks. However, convergence problem has been observed for the medium size networks (networks $N_3$ and $N_4$) and large size networks (networks $N_5$ and $N_6$). It is worth noting that the relatively head differences use in the stopping test has a better convergence property that the continuity residual. This is particularly pronounced in network $N_2$ as shown in Figure 2b. The stopping test has been met at seven iteration, whereas the continuity residual is $10^{-3}$. This significant difference is also observed in Figure 2c–f for networks $N_3$–$N_6$.

Using junction 776 and reservoir E in network $N_3$ as an example, the head iterates of junction 776 is observed to be oscillating around the elevation head of reservoir E as shown in Figure 3a. The relative head difference of the nodal head at junction 775 is $3.94 \times 10^{-10}$ at 6th iteration. However, this small perturbation in the nodal head caused the flow direction reversed. This flow direction reversal happened three more times while the value of the nodal heads converges to the true value. This is because the pipe (Pipe 1865) that connects junction 776 and reservoir E is 1 m in length and 0.999 m in diameter with 0.15 mm roughness height, which means the resistance factor of pipe 1865 is a very small value. This can also be seen from Figure 3a as the head loss between reservoir E and junction 776 is $10^{-8}$.

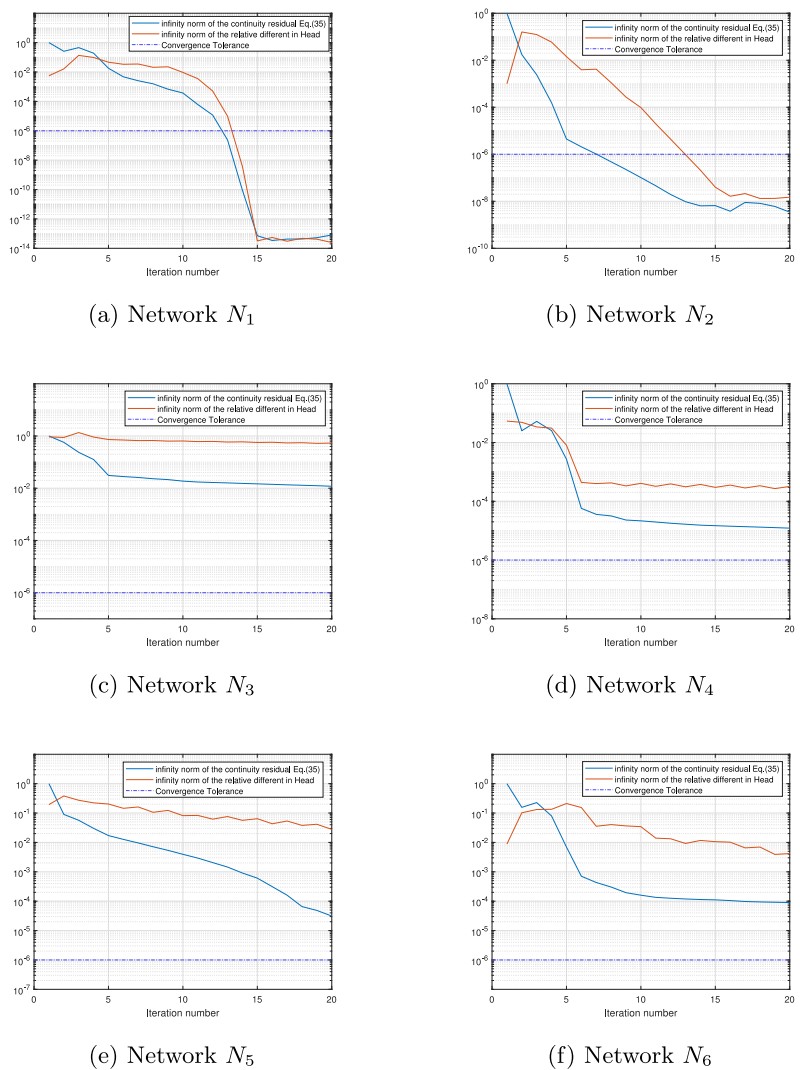

(a) Network $N_1$      (b) Network $N_2$

(c) Network $N_3$      (d) Network $N_4$

(e) Network $N_5$      (f) Network $N_6$

**Figure 2.** The infinity norm of the relative head difference $\left\| \dfrac{h^{m+1} - h^m}{h^{m+1}} \right\|_\infty$ and infinity norm of the continuity residuals after applied the proposed head formulation to each of the six case study networks $N_1$–$N_6$ using the Newton method as shown in Figure 2, at sub-figures (**e**–**f**).

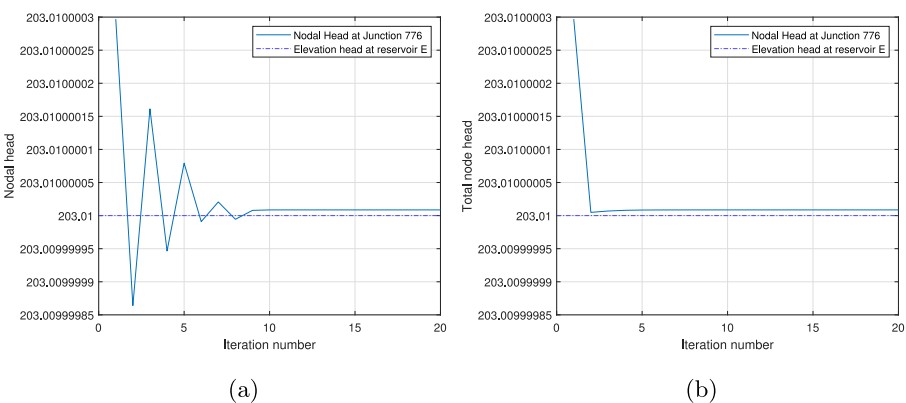

(a)      (b)

**Figure 3.** The convergence property of the proposed head formulation applied to the Balerma Network: (**a**) the head iterates at Junction 776 in Balerma Network and source elevation at Reservoir E using the Newton method; (**b**) the head iterates at Junction 776 in Balerma Network and source elevation at Reservoir E using the damped Newton method.

Once a damping factor of 0.67 has been applied, the head convergence of this particular node is more well-behaved as a faster convergence is achieved and no head oscillation has been observed as can be seen from Figure 3b. Please note that in Figure 3b and in Figure 4 the damped Newton's method was used. Within this scheme, the derivative within the Newton method is multiplied by a damping factor between zero and one for accelerating convergence. On the one hand, the application of the damping factor caused a slower head convergence of the node that is well-behaved before its application. This can be seen from Figure 4a,b as a slower rate of convergence is observed when compared to that from Figure 2a,b. On the other hand, the application of the damping factor guaranteed convergence for networks $N_3$–$N_6$ as shown in Figure 4c–f when compared to before the application of the damping factor as shown in Figure 2c–f.

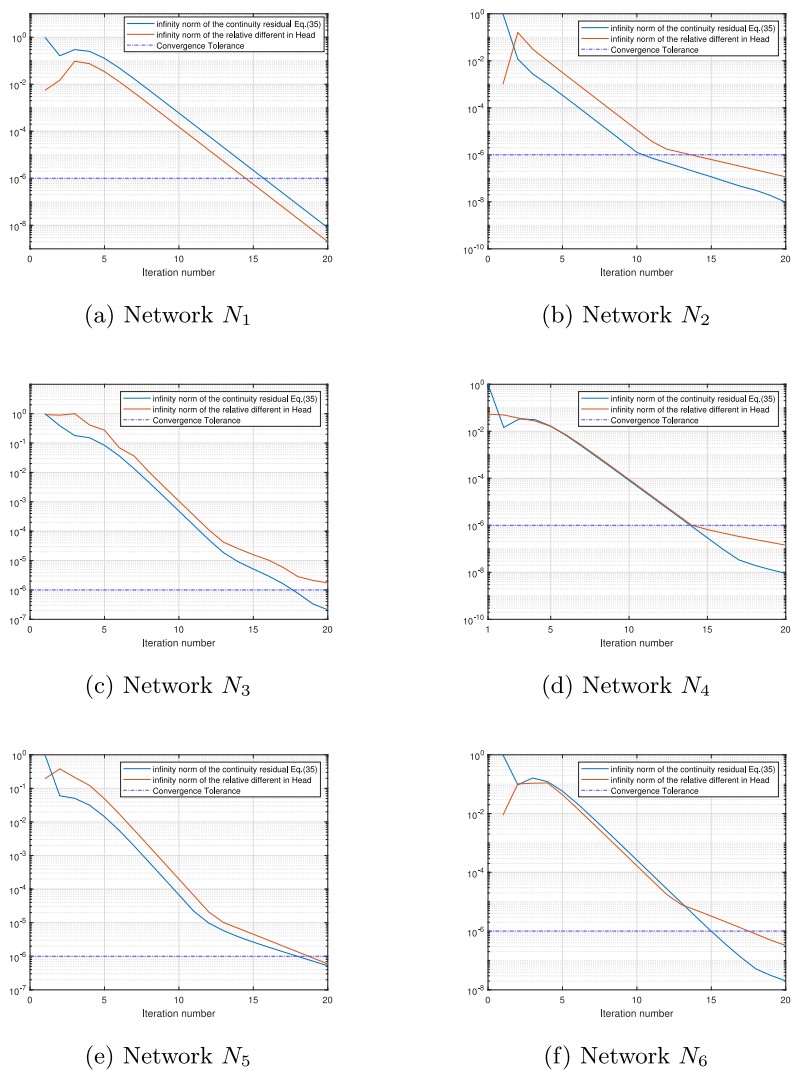

(a) Network $N_1$ (b) Network $N_2$

(c) Network $N_3$ (d) Network $N_4$

(e) Network $N_5$ (f) Network $N_6$

**Figure 4.** The infinity norm of the relative head difference $\left\lVert \dfrac{h^{m+1} - h^m}{h^{m+1}} \right\rVert_\infty$ and infinity norm of the continuity residuals after applied the proposed head formulation to each of the six case study networks $N_1$–$N_6$ using the damped Newton method as shown in Figure 4, at sub-figures (**e**–**f**).

The convergence properties, including the relative head differences, continuity residuals, energy residuals, and Colebrook–White equation residuals, at the final iteration of the proposed head formulation after applying the damping factor is shown in Table 2.

**Table 2.** Convergence property of the proposed head formulation applied to the six case study networks.

| Network | Num. of Iter. | $\left\|\left\|\frac{h^{m+1}-h^{m+1}}{h^{m+1}}\right\|\right\|_\infty$ | $\|\sigma_C\|_\infty$ | $\|\sigma_E\|_\infty$ | $\|\sigma_f\|_\infty$ |
|---------|---------------|------|------|------|------|
| $N_1$ | 16 | $7 \times 10^{-7}$ | $1 \times 10^{-7}$ | $1 \times 10^{-15}$ | $1 \times 10^{-15}$ |
| $N_2$ | 11 | $7 \times 10^{-7}$ | $9 \times 10^{-6}$ | $1 \times 10^{-15}$ | $2 \times 10^{-15}$ |
| $N_3$ | 18 | $7 \times 10^{-7}$ | $2 \times 10^{-6}$ | $1 \times 10^{-15}$ | $1 \times 10^{-15}$ |
| $N_4$ | 14 | $9 \times 10^{-7}$ | $5 \times 10^{-6}$ | $1 \times 10^{-15}$ | $1 \times 10^{-15}$ |
| $N_5$ | 18 | $9 \times 10^{-7}$ | $1 \times 10^{-6}$ | $1 \times 10^{-15}$ | $1 \times 10^{-15}$ |
| $N_6$ | 15 | $9 \times 10^{-7}$ | $2 \times 10^{-6}$ | $1 \times 10^{-15}$ | $3 \times 10^{-15}$ |

Table 3 presents the detailed timing results of pre-processing, iterative phase and post-processing operations of the GGA and proposed head formulation applied to the six case study networks. The total wall-clock time required to apply the proposed head formulation is higher than that required by the GGA for all six case study networks. This is because the proposed head formulation requires more time to perform the iterative phase while the GGA and the proposed head formulation require a similar time to execute the pre-processing and post-processing operations. In addition, a similar amount of per-iteration runtime is required by the GGA and the proposed head formulation, which means the longer wall-clock time required by the proposed head formulation is the result of higher number of iterations required due the damping factor applied.

**Table 3.** The number of iterations and the wall-clock time (second) required to perform the pre-processing, iterative phase, and the post-processing operations for the GGA and the proposed head formulation applied to each of the six case study networks (the number in the bracket indicates the per-iteration run time required to execute the iterative phase of the Newton method).

| Network | Methods | Num. of Iter. | Preproc. | Iterative Phase | PostProc. | Total |
|---------|---------|---------------|----------|-----------------|-----------|-------|
| $N_1$ | GGA | 6 | 0.003 | 0.003 (0.0005) | 0.001 | 0.007 |
|        | Head | 16 | 0.004 | 0.009 (0.0005) | 0.001 | 0.014 |
| $N_2$ | GGA | 6 | 0.005 | 0.006 (0.001) | 0.001 | 0.012 |
|        | Head | 11 | 0.007 | 0.013 (0.001) | 0.001 | 0.021 |
| $N_3$ | GGA | 9 | 0.011 | 0.025 (0.003) | 0.001 | 0.037 |
|        | Head | 18 | 0.014 | 0.053 (0.003) | 0.001 | 0.068 |
| $N_4$ | GGA | 6 | 0.032 | 0.027 (0.005) | 0.001 | 0.060 |
|        | Head | 14 | 0.024 | 0.070 (0.005) | 0.001 | 0.095 |
| $N_5$ | GGA | 8 | 0.062 | 0.143 (0.018) | 0.002 | 0.207 |
|        | Head | 18 | 0.089 | 0.367 (0.02) | 0.001 | 0.457 |
| $N_6$ | GGA | 8 | 1.93 | 1.78 (0.22) | 0.02 | 3.73 |
|        | Head | 15 | 2.14 | 3.61 (0.26) | 0.006 | 5.75 |

Table 4 shows the detailed statistics of the absolute head differences between the GGA and the proposed head formulation applied to each of the six case study networks. This head differences in the results of optimization problems can cause constraints violation of the optimal solution identified. Take the Balerma network $N_1$, one of the well explored network, as an example. This problem is first introduced by [17]. Two least-cost designs presented by [23,24], both of which using the GGA formulation to find the steady-state solution in which the Swamee–Jain equation is used to model the pipe friction factors, have found to have nodal pressure violations when using the proposed head formulation as shown in Table 5, in which the Colebrook–White equation is used to model the pipe friction factors. The above two least-cost designs of network $N_3$ are both infeasible as the

Swamee–Jain equation approximates the Colebrook–White equation. This is because (1) the errors involved in Swamee–Jain friction factor are within 1% for $\epsilon/D \in [10^{-6}, 10^{-2}]$ and $Re \in [5000, 10^8]$ as reported in [13], whereas GGA has extended the applicability of the Swamee–Jain equation to all values of $\epsilon/D$ and $Re \geq 4000$ and (2) the errors involved in Swamee–Jain friction factor have been ignored in all optimization problems.

**Table 4.** The detailed statistics of the absolute head differences ($m$) between the GGA and the proposed head formulation, $|h_{GGA} - h_{head}|$, applied to six case study networks.

| Network | Min | Mean | Median | Max | std.dev |
|---------|-----|------|--------|-----|---------|
| $N_1$ | 1.00e-4 | 8.98e-2 | 7.89e-2 | 3.07e-1 | 5.99e-2 |
| $N_2$ | 0.00 | 3.12e-3 | 2.70e-3 | 9.90e-3 | 2.14e-3 |
| $N_3$ | 5.00e-4 | 2.04e-1 | 2.12e-1 | 3.09e-1 | 5.27e-2 |
| $N_4$ | 1.30e-2 | 1.20e-1 | 1.19e-1 | 2.37e-1 | 5.13e-2 |
| $N_5$ | 4.65e-2 | 5.88e-2 | 5.88e-2 | 8.15e-2 | 3.72e-3 |
| $N_6$ | 3.00e-3 | 2.31e-2 | 1.61e-2 | 9.22e-2 | 1.62e-2 |

**Table 5.** The nodal pressure violation in some of the solutions of least-cost design of network $N_1$ using the proposed head formulation.

| (a) €1,940,923 design found in Tolson et al. [23] | | | |
|-----------------|-----------|------------|------------------|
| Junction | Elevation | Total Head | Pressure Deficit |
| Junc 3 | 23.90 | 43.62 | 19.72 | 2.76e-1 |
| Junc 59 | 21.00 | 40.92 | 19.92 | 8.18e-2 |
| Junc 151 | 56.50 | 76.39 | 19.89 | 1.14e-1 |
| Junc 233 | 87.17 | 107.08 | 19.91 | 9.08e-2 |
| Junc 270 | 73.70 | 93.59 | 19.89 | 1.07e-1 |
| Junc 281 | 75.00 | 94.92 | 19.92 | 8.46e-2 |
| Junc 332 | 75.70 | 95.60 | 19.90 | 9.69e-2 |
| Junc 359 | 80.70 | 100.62 | 19.92 | 8.25e-2 |
| Junc 363 | 80.50 | 100.41 | 19.91 | 9.23e-2 |
| Junc 394 | 56.40 | 76.30 | 19.90 | 1.01e-1 |
| Junc 397 | 80.50 | 100.41 | 19.91 | 8.62e-2 |
| Junc 398 | 80.50 | 100.40 | 19.90 | 9.84e-2 |
| Junc 401 | 81.70 | 101.70 | 19.99 | 9.00e-4 |

| (b) €1,920,656 design found in Barlow and Tanyimboh [24] | | | | |
|-----------------|-----------|------------|---------------|------------------|
| Junction | Elevation | Total Head | Pressure Head | Pressure Deficit |
| Junc 3 | 23.90 | 43.72 | 19.82 | 1.81e-1 |
| Junc 135 | 60.00 | 79.95 | 19.95 | 5.12e-2 |
| Junc 150 | 45.00 | 64.82 | 19.82 | 1.76e-1 |
| Junc 152 | 55.00 | 74.99 | 19.99 | 2.38e-2 |
| Junc 201 | 95.00 | 114.95 | 19.95 | 5.03e-2 |
| Junc 233 | 87.17 | 107.11 | 19.94 | 5.58e-2 |
| Junc 281 | 75.00 | 94.96 | 19.96 | 4.09e-2 |
| Junc 359 | 80.70 | 100.59 | 19.89 | 1.07e-1 |
| Junc 363 | 80.50 | 100.41 | 19.91 | 8.83e-2 |
| Junc 374 | 69.50 | 89.41 | 19.91 | 8.75e-2 |
| Junc 397 | 80.50 | 100.41 | 19.91 | 9.10e-2 |
| Junc 398 | 80.50 | 100.39 | 19.89 | 1.12e-1 |
| Junc 401 | 81.79 | 101.69 | 19.89 | 5.70e-3 |
| Junc 179,001 | 60.00 | 79.96 | 19.96 | 4.00e-2 |

In addition to the differences in the head solutions, the use of friction factor produced by the Colebrook–White equation in the proposed head formulation also produces different

flow results when compared to that produced by the GGA as shown in Table 6 and the spatial distribution of the pipes with different flows is shown in Figure 5. The flows of the pipes in the looped component found by using the proposed head formulation for each of the six case study networks are different from that found by using the GGA, whereas the flows of the pipes in the forest component found by using the head formulation for each of the six case study networks are the same as that found by using the GGA. In addition, the number of pipes with flow direction reversal is ranging from six pipes in network $N_2$ to 730 pipes in network $N_6$.

In addition to the single-period steady-state WDS simulation where the boundary conditions (pumps and tanks) are fixed, errors caused by the difference between the Colebrook–White equation and the Swamee–Jain equation can accumulate as the boundary condition in the extended-period simulation. The error accumulation is manifested as the different tank level at each time step, the time when an operation activated by the trigger level, and the pump operating points. Net3 in EPANET is used as an example to demonstrate the error accumulation described above. It is important to note that the Net3, which is a simple WDS with 92 nodes, two reservoirs, three tanks, two pumps, and 117 pipes, in EPANET has been converted from the Hazen-William head loss model to the Darcy–Weisbach head loss model. The model difference between the GGA and the head formulation is relatively small between 00:00 to 05:00, which is also observed in network $N_1$. As can be seen from Figure 6, however, the tank levels start to diverge as the tank level rises. This is because the insignificant model errors for small network can accumulate over a period of time in the form of boundary conditions. It is also worth noting that when the tank operation mode changes, from filling to discharging or from discharging to filling, the tank level differences between the two model start to narrowing, because the cancelling of errors, but never reaching zero. The maximum tank level difference is 0.074 m for tank 1, 0.12 m for tank 2 and 0.37 m for tank 3. Due to the error build-up in the boundary conditions, the differences between GGA and the head formulation flow results are relatively small with no flow direction reversal before the mid-day as can be seen from Figure 7a. After 12:00, however, significant differences between GGA and the head formulation flow results are observed with the reversal of flow direction as can be seen from Figure 7c–e. Finally, flow result differences start to decrease as the tank level differences narrow. The GGA and the head formulation start with the same tank levels and end up with different tank levels over the 24 h simulation period. This boundary condition differences will keep accumulating over a longer simulation period.

**Table 6.** The number of pipes in different bins of the relative difference between the Flow results of the GGA and the proposed head formulation applied to six case study networks.

| | $N_1$ | $N_2$ | $N_3$ | $N_4$ | $N_5$ | $N_6$ |
|---|---|---|---|---|---|---|
| Same | 302 (66.52%) | 371 (39.72%) | 438 (17.77%) | 1567 (38.97%) | 2951 (19.90%) | 45,788 (29.16%) |
| 0–1% | 146 (32.16%) | 390 (41.76%) | 858 (34.81%) | 2312 (57.50%) | 8113 (54.71%) | 93,859 (59.77%) |
| 1–10% | (1.32%) | 138 (14.78%) | 671 (27.22%) | 106 (2.64%) | 1926 (12.99%) | 11,210 (7.14%) |
| 10–20% | 0 (0.00%) | 15 (1.61%) | 193 (7.83%) | 4 (0.10%) | 295 (1.99%) | 1684 (1.07%) |
| 20–30% | 0 (0.00%) | 2 (0.21%) | 85 (3.45%) | 8 (0.20%) | 179 (1.21%) | 940 (0.60%) |
| 30–40% | 0 (0.00%) | 2 (0.21%) | 43 (1.74%) | 1 (0.02%) | 125 (0.84%) | 648 (0.41%) |
| 40–50% | 0 (0.00%) | 0 (0%) | 26 (1.05%) | 0 (0.00%) | 117 (0.79%) | 453 (0.29%) |
| >50% | 0 (0.00%) | 16 (1.71%) | 151 (6.13%) | 23 (0.57%) | 1124 (7.58%) | 2462 (1.57%) |
| Flow reversal | 0 (0.00%) | 7 (0.75%) | 45 (1.83%) | 12 (0.30%) | 448 (3.02%) | 730 (0.46%) |

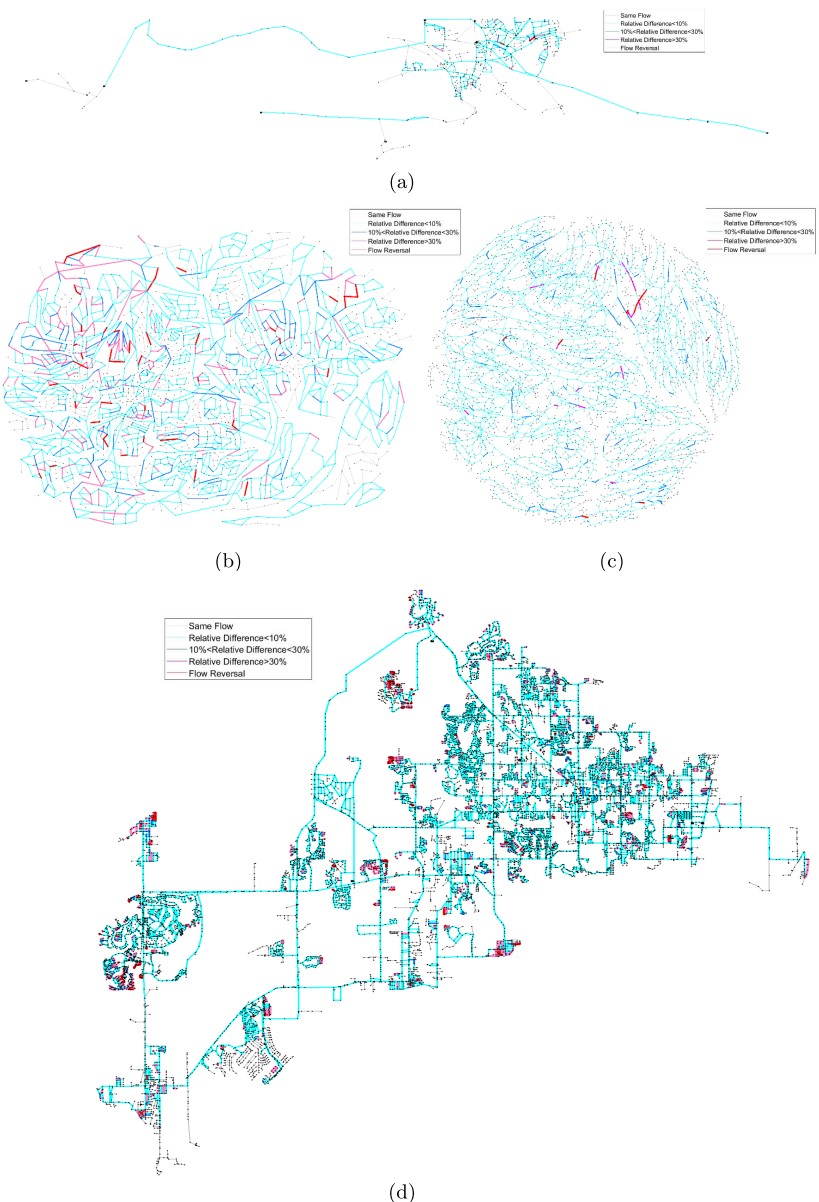

**Figure 5.** The spatial distribution of the different levels of the relative differences between flow results of the GGA and the proposed head formulation applied to (**a**) network $N_2$, (**b**) network $N_3$, (**c**) network $N_4$, and (**d**) network $N_5$.

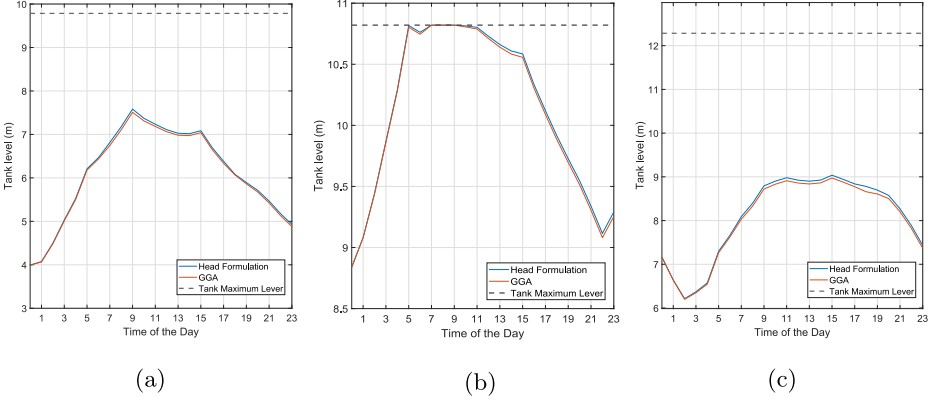

**Figure 6.** The differences between the tank level found by applying the head formulation and the GGA to the network Net3 in EPANET for the network three tanks, as shown in sub-figures (**a**–**c**).

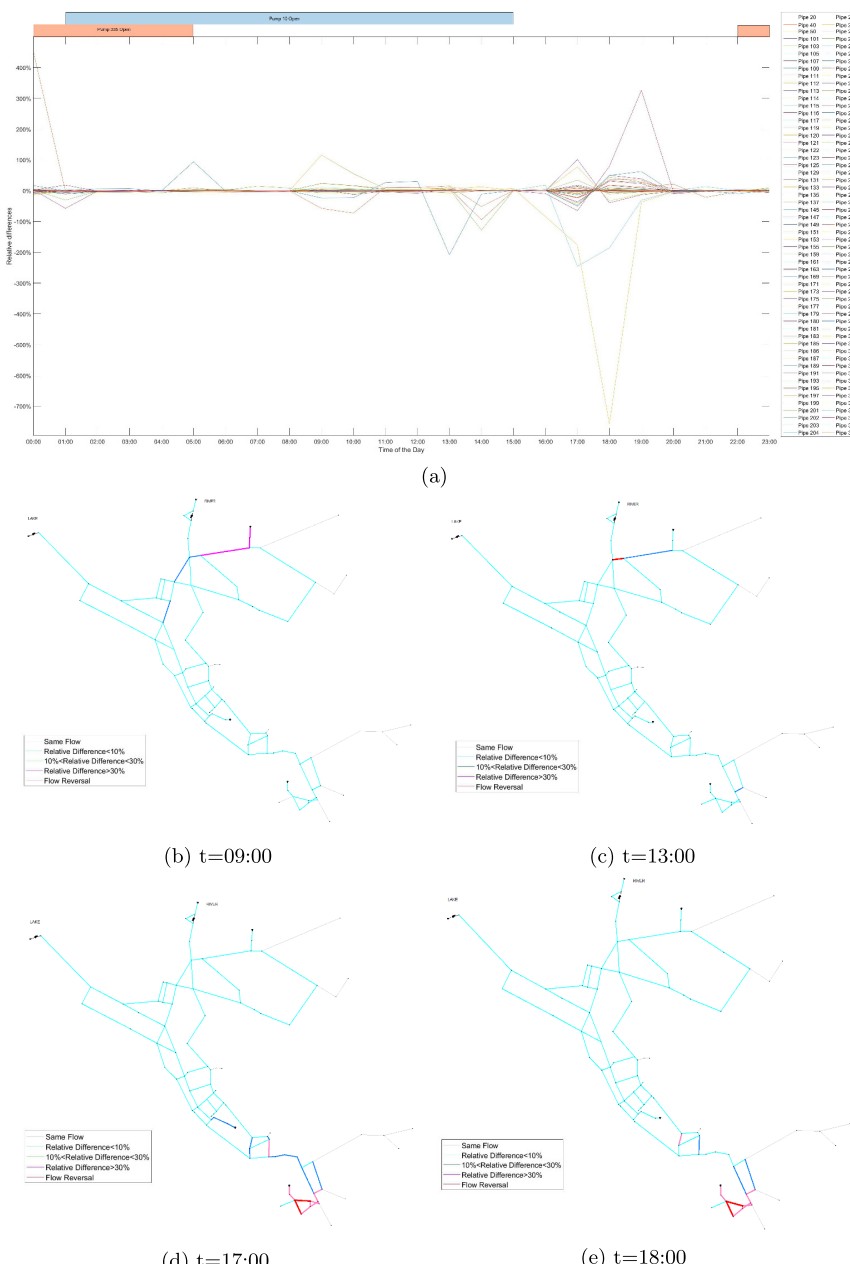

(a)

(b) t=09:00

(c) t=13:00

(d) t=17:00

(e) t=18:00

**Figure 7.** The relative differences between the flows in each pipe of the EPANET example 3 found by the GGA and the proposed head formulation over 24 h extended-period simulation: (**a**) temporal variation over the 24 h extended-period simulation, (**b**) spatial distribution of different levels of relative difference of pipe flows at *t* = 09:00, (**c**) spatial distribution of different levels of relative difference of pipe flows at *t* = 13:00, (**d**) spatial distribution of different levels of relative difference of pipe flows at *t* = 17:00, and , (**e**) spatial distribution of different level of relative difference of pipe flows at *t* = 18:00.

## 8. Possible Applications

The main improvement of using the proposed head formulation is the use of the Colebrook–White equation addresses a fundamental error that is associated with the use of Swamee–Jain equation, an approximation of the Colebrook–White equation, in all existing WDS solution methods. It is shown in the last section that both flow and head results of the head formulation are different from that of the global gradient algorithm. The differences in flow and head results can significantly affect several research areas.

### 8.1. Water Quality Simulation

Water quality is one of the most important water distribution system research subject. Water quality is simulated using the chemical transportation equation (advection) and the chemical decay equation, both of which are functions of pipe flows and time for both the contaminant and the disinfectant. As a result, the differences in flow results between the proposed head formulation and the GGA, particularly the flow reversal, can significantly affect the evaluation of the chemical transportation equation. This can be a critical problem for the identification of the contamination sources, network decontamination, disinfection by-product modeling, water quality sensor placement, and others.

In addition, the water quality modeling is temporal in nature, therefore, the flow results of the extended-period simulation are normally used in the chemical transportation equation. As can be seen from Figure 7 that the differences in flow results have no correlation and the pipes with flow reversal are not always the same. As a result, there is another level of error accumulation in water quality simulation on top of the extended-period simulation as opposed to the water quality simulation using the steady-state simulation where only the initial conditions are different. These two levels of error accumulation are compounded so that the errors in the water quality modeling can be amplified.

### 8.2. Water Hammer Analysis

Water hammer analysis is another research area that is affected by the flows, heads, and head losses in pipes and nodes of water distribution systems. The governing equations for the water hammer analysis are the unsteady momentum equation and the unsteady continuity equation. Both unsteady equations are functions of heads, flows, and the direction of flows. The reversal of the flow direction in a pipe will significantly alter the result of water hammer analysis in a looped water distribution system.

### 8.3. WDS Network Calibration

The proposed head formulation can also improve the quality of the WDS calibration models. Water distribution system calibration is a process of comparing model results with field data and making the appropriate adjustments so that both results agree, and it is usually applied to the estimation of pipe roughness values and nodal demands [25]. All existing calibration studies compared the *'measurement data'*, which is generated using EPANET instead of using the real measurement data, against the existing model results, both of which uses the Swamee–Jain equation to model the turbulent flow regime. As a result, the errors existed in the Swamee–Jain approximation is inherited by the calibrated water distribution system. A better network calibration model can be produced by using the proposed head formulation as the differences between the Colebrook–White equation and the experimental data are smaller than that between the Swamee–Jain equation and the experimental data.

### 8.4. WDS Optimization and Operation

The small head differences can invalidate all existing optimal solutions of water distribution system modeled by Darcy–Weisbach head loss equation. Examples of the pressure violations in the optimal solutions found in [23,24] for the Balerma network using the global gradient algorithm using Swamee–Jain equation are shown in Table 5, both of which become infeasible if the proposed Colebrook–White equation is used.

## 9. Conclusions

This paper presents an efficient iterative head formulation for the steady-state demand-driven solution of water distribution systems for both Darcy–Weisbach and Hazen-Williams head loss model. When the Hazen-Williams head loss model is used, the proposed solution method produces the same final and iterative flow and head solutions if the same initial guess is used as the global gradient algorithm. When the Darcy–Weisbach head loss model is used, an exact and explicit expression of the inexplicit Colebrook–White friction factor

equation is proposed in this study. A cubic interpolation between this explicit expression of the inexplicit Colebrook-White equation for the turbulent flow regime and the Hagen-Poiseuille equation for the laminar flow regime is generated to describe the friction factor in the transitional flow regime.

The main features of the proposed head formulation of the steady-state demand-driven WDS simulation include:

1.  friction factor for the turbulent flow regime can be calculated using an explicit and exact expression of the Colebrook–White equation without the need for an iterative method;

2.  the use of the proposed head formulation can significantly improve the accuracy of the steady-state demand-driven solution of WDSs when compared to the GGA. This is because the proposed head formulation uses an explicit and exact expression of the Colebrook–White equation to calculate friction factor for the turbulent flow regime as opposed to the Swamee–Jain equation, an approximation of the Colebrook–White equation, used in previous WDS solution methods.

The efficacy of the proposed head formulation has been demonstrated by applying it to six case study networks, the results of which have been validated using the continuity residuals, energy residuals, and Colebrook–White residuals. It should also be observed that the proposed method could be selected for analytical (e.g., Di Nucci [26]) and for numerical applications (e.g., Pasculli [27]), and in particular for improving the Wall Function.Differences between the proposed head formulation and the GGA have been observed in both the flows and heads. On the one hand, the flow differences, particularly the flow direction reversal, can be a critical problem for some research areas such as water quality simulation, contamination source identification, water hammer analysis, WDS network calibration, sensor placement, and network clustering. On the other hand, the heads differences can cause pressure violations in the WDS least-cost design when the GGA is used to perform the steady-state demand-driven analysis. It is important to note that damping factor has been applied to the Newton method to achieve a more stable convergence. The choice of initial guess, another important factor in the Newton method convergence, and inclusion of valves, pumps, and other network elements will be interesting avenues for future research.

**Author Contributions:** Conceptualization, M.Q.; Data curation, M.Q.; Formal analysis, M.Q.; Funding acquisition, A.O.; Methodology, M.Q.; Project administration, A.O.; Supervision, A.O.; Validation, M.Q.; Visualization, M.Q.; Writing—original draft, M.Q.; Writing—review and editing, M.Q. and A.O. All authors have read and agreed to the published version of the manuscript.

**Funding:** This research was funded by the ISRAEL SCIENCE FOUNDATION (grant No. 555/18).

**Institutional Review Board Statement:** Not applicable.

**Informed Consent Statement:** Informed consent was obtained from all subjects involved in the study.

**Data Availability Statement:** The data presented in this study are available on request from the corresponding author.

**Acknowledgments:** This research was supported by the ISRAEL SCIENCE FOUNDATION (grant No. 555/18).

**Conflicts of Interest:** The authors declare no conflict of interest.

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
