# Peer review of "A Head Formulation for the Steady-State Analysis of Water Distribution Systems Using an Explicit and Exact Expression of the Colebrook–White Equation"

_water, doi:10.3390/w13091163_

Round 1

Reviewer 1 Report

This manuscript presents a very interesting work on the steady-state analysis of water distribution systems. A formulation is presented for the solution of WDSs using the Colebrook-White equation. The correctness and efficacy of the head formulation were demonstrated by applying it to various WDSs. The writing in this study is good. The research work and related conclusions are worthful. I suggest that the paper is accepted.

Author Response

Thanks a lot for your comments. Much appreciated.

Reviewer 2 Report

Manuscript is prepared in poor form. It should be prepared according to the journals guidelines. The manuscript presents the well-known method. Link to novelty is weak. Lot of literature about presented issue have been done.

Author Response

We have addressed all reviewers’ comments and complied with the journal format. We hope that the paper is now improved and can be accepted for publication.

Reviewer 3 Report

please find the attached file

Author Response

Comment 1

The article discusses an effective method for calculating the friction factor that allows for the easy application of the exact expression of the Colebrook-White equation. The paper is well presented and enjoyable to read.

Response 1

Thank you.

Comment 2

However, to complete the content it would be desirable that the authors include more details about the application of the Damping Newton Rapshon Method, selected to carry out the calculations.

Response 2

Added to the manuscript more details on the Damping Newton Rapshon Method, selected to carry out the calculations: “Note that in Fig. 3b and in Fig. 4 the damped Newton’s method was utilized. Within this scheme, the derivative within the Newton method is multiplied by a damping factor between zero and one for accelerating convergence.”

Comment 3

Furthermore, it would be useful that they observe that their method could be usefully selected both from the analytical point of view, for example:

Di Nucci, C., Petrilli M., Russo Spena, A. (2011). Unsteady friction and viscoelasticity in pipe fluid transients. J. Hydraul. Res. 49 (3), and from the numerical point of view, for example: Pasculli, A. (2008). CFD-FEM 2D Modelling of a local water flow. Some numerical results. Alpine and Mediterranean Quaternary, 21(B), Issue 1, 2008, 215-228. In particular for the improvement of the so-called Wall Functions

Response 3

Added clarifications on the method usefulness applicability to be selected both from the analytical point of view, and from the numerical point of view. Added also the two suggested references. We added to the conclusions section: “It should also be observed that the proposed method could be selected for analytical (e.g., Di. Nucci et al., 2017) and for numerical applications (e.g., Pasculli, 2008), and in particular for improving the Wall Function.”

Reviewer 4 Report

see the enclosed file

Author Response

(The authors gave the same response as above.)
